# Walk Your Talk: Real-World Adherence to Guidelines on the Use of MRI in Multiple Sclerosis

**DOI:** 10.3390/diagnostics11081310

**Published:** 2021-07-21

**Authors:** Mario Tortora, Mario Tranfa, Anna Chiara D’Elia, Giuseppe Pontillo, Maria Petracca, Alessandro Bozzao, Ferdinando Caranci, Amedeo Cervo, Mirco Cosottini, Andrea Falini, Marcello Longo, Renzo Manara, Mario Muto, Michele Porcu, Luca Roccatagliata, Alessandra Todeschini, Luca Saba, Arturo Brunetti, Sirio Cocozza, Andrea Elefante

**Affiliations:** 1Department of Advanced Biomedical Sciences, University of Naples “Federico II”, 80131 Naples, Italy; mario.tortora@ymail.com (M.T.); mariotranfa@libero.it (M.T.); deliaannachiara@gmail.com (A.C.D.); giuseppe.pon@gmail.com (G.P.); brunetti@unina.it (A.B.); aelefant@unina.it (A.E.); 2Department of Neurosciences and Reproductive and Odontostomatological Sciences, University “Federico II”, 80131 Naples, Italy; maria.petracca@unina.it; 3Department of Human Neurosciences, Sapienza University of Rome, 00189 Rome, Italy; 4Neuroradiology Unit, NESMOS Department, Sapienza University of Rome, 00189 Rome, Italy; alessandro.bozzao@uniroma1.it; 5Department of Medicine of Precision, University of Campania “Luigi Vanvitelli”, 80138 Naples, Italy; ferdinando.caranci@unicampania.it; 6Department of Neuroradiology, ASST Grande Ospedale Metropolitano Niguarda, 20121 Milan, Italy; amedeo.cervo@gmail.com; 7Department of Translational Research and New Technologies in Medicine and Surgery, University of Pisa, 56126 Pisa, Italy; mirco.cosottini@unipi.it; 8Neuroradiology Department, IRCCS San Raffaele Hospital and University, 20132 Milan, Italy; falini.andrea@unisr.it; 9Neuroradiology Unit, Department of Biomedical Sciences and Morphological and Functional Images, University of Messina, 98124 Messina, Italy; mlongo@unime.it; 10Department of Neurosciences, University of Padua, 35121 Padua, Italy; renzo.manara@unipd.it; 11Diagnostic and Interventional Neuroradiology, Cardarelli Hospital, 80131 Naples, Italy; mario.muto@aocardarelli.it; 12Department of Radiology, Azienda Ospedaliero Universitaria (A.O.U.) di Cagliari, 09124 Cagliari, Italy; micheleporcu87@gmail.com (M.P.); lucasabamd@gmail.com (L.S.); 13Department of Health Sciences, University of Genova, 16132 Genova, Italy; lroccatagliata@neurologia.unige.it; 14Neuroradiology Unit, IRCCS Ospedale Policlinico San Martino, 16132 Genova, Italy; 15Neuroradiology Unit, Department of Neuroscience, Nuovo Ospedale Civile S. Agostino Estense, 41126 Modena, Italy; todeschini.ale@gmail.com

**Keywords:** MRI, multiple sclerosis, neuroradiology

## Abstract

(1) Although guidelines about the use of MRI sequences for Multiple Sclerosis (MS) diagnosis and follow-up are available, variability in acquisition protocols is not uncommon in everyday clinical practice. The aim of this study was to evaluate the real-world application of MS imaging guidelines in different settings to clarify the level of adherence to these guidelines. (2) Via an on-line anonymous survey, neuroradiologists (NR) were asked about MRI protocols and parameters routinely acquired when MS patients are evaluated in their center, both at diagnosis and follow-up. Furthermore, data about report content and personal opinions about emerging neuroimaging markers were also retrieved. (3) A total of 46 participants were included, mostly working in a hospital or university hospital (80.4%) and with more than 10 years of experience (47.9%). We found a relatively good adherence to the suggested MRI protocols regarding the use of T2-weighted sequences, although almost 10% of the participants routinely acquired 2D sequences with a slice thickness superior to 3 mm. On the other hand, a wider degree of heterogeneity was found regarding gadolinium administration, almost routinely performed at follow-up examination (87.0% of cases) in contrast with the current guidelines, as well as a low use of a standardized reporting system (17.4% of cases). (4) Although the MS community is getting closer to a standardization of MRI protocols, there is still a relatively wide heterogeneity among NR, with particular reference to contrast administration, which must be overcome to guarantee an adequate quality of patients’ care in MS.

## 1. Introduction

Multiple sclerosis (MS) is an autoimmune inflammatory disease affecting the central nervous system (CNS), leading to white matter (WM) demyelination, gray matter (GM) atrophy, and global neurodegeneration [1]. It shows a higher incidence in young adults, especially women, and it is characterized by a heterogeneous spectrum of symptoms and clinical phenotypes, which correspond to a variable degree of neurological disability [2]. Given this relatively wide heterogeneity in clinical presentation, there are many pathological entities that have to be considered and excluded in MS differential diagnosis [3]. The role of Magnetic Resonance Imaging (MRI) in supporting MS diagnosis has been clearly established over the past years. Indeed, the McDonald diagnostic criteria (from their first draft in 2001 to their latest revision in 2017) [4] are based on the demonstration of dissemination in space (DIS) and time (DIT) using MRI or through clinical symptoms and cerebrospinal fluid (CSF) markers, respectively. However, not only from a clinical but also from an MRI perspective, several conditions can mimic MS [5]. In this light, in order to facilitate the application of the McDonalds diagnostic criteria to avoiding misdiagnosis and to favor the MRI monitoring of disease activity over time [6], specific imaging protocols have been developed and are periodically revised and refined by experts in the field [7,8]. In Europe, a central role is played by the Magnetic Resonance Imaging in MS (MAGNIMS) guidelines, drafted by a network of MS experts [9]. Unfortunately, even though the MAGNIMS consensus guidelines clearly define the MRI sequences mandatory for both baseline and follow-up examinations [6,10], variability in MRI acquisition protocols is not an uncommon event in everyday clinical practice. In order to clarify the level of adherence to published guidelines [9] in clinical settings, we developed an anonymous online survey. Neuroradiologists (NR) working in different and relatively heterogeneous settings (ranging from university hospitals to private medical practice) were asked about the protocols applied for MS diagnosis/monitoring and about the study report produced for clinical purposes, which ideally should contain all the information necessary to allow diagnosis and monitor response to therapy.

## 2. Materials and Methods

For the dissemination of the questionnaire, based on our connections, we reached at least three neuroradiologists (one for each setting explored) per each Italian region in an attempt to enroll a representative sample. We then asked each reached NR who showed interest in participating in this work to also disseminate the survey to colleagues working in different centers, with the aim of increasing the sample size. The questionnaire was published online via the EUSurvey website (https://ec.europa.eu/eusurvey accessed on 1 May 2021), and each participant was asked to respond according to their real-life experience rather than their theoretical knowledge of the current guidelines. The questionnaire, redacted in Italian, was fully anonymous and designed to be completed in 10 to 15 min, presenting different sets of questions regarding the participant’s professional profile, the MRI protocols used for diagnosis and follow up of MS, and the format and content of the study report. The complete questionnaire, translated from Italian to English, is available in Appendix A. Briefly, the questionnaire was created to first obtain information about the participant’s professional background (e.g., by retrieving information about the respondent’s level of experience or where he/she usually performs the neuroradiological evaluation). A second set of questions covered the MRI protocol used for MS diagnosis, including the field strength used to evaluate these patients, the type of sequences used, and some additional information about their spatial resolution or time relation with gadolinium administration. The same questions were then asked regarding the neuroradiological practice at the follow-up examination. After the questions regarding the technical details of the acquisition, information about the neuroradiological report were retrieved. Finally, NRs were asked their opinion about some possible future directions of clinical neuroimaging in MS, such as the use of a new emerging sign or the application of more quantitative biomarkers of disease, such as the inclusion of an automatically generated report of the degree of brain atrophy. The Survey was published online for approximately 3 months from 9 February 2021. On 19 April 2021, the enrollment phase was considered closed. Survey responses were then retrieved end exported from the website as a.csv file, from which descriptive statistics were calculated. Results are presented for the entire group of participants, as well as stratified by years of experience and workplace (baseline acquisition only), in order to evaluate possible differences in terms of application depending on the work environment.

## 3. Results

### 3.1. Participants

Forty-six NRs completed the online survey, including 2 residents (4.3%), 18 young NRs (39.1%) (defined as board-certified NRs with less than 5 years of experience), 4 NRs (8.7%) (defined as board-certified NRs with more than 5 years but less than 10 years of experience), and 22 NRs (47.9%) with more than 10 years of experience. A large proportion of participants worked in a hospital or university hospital (37/46, 80.4%), with the remaining 9 NRs (19.6%) evaluating MS patients in private clinical practice. More than 50% of the respondents (52.2%) declared that they evaluate, on average, more than 10 MS cases per month, in most cases (67.4%) using a 1.5 T scanner. Information about participants’ experience and work setting are displayed in Figure 1.

### 3.2. Brain and Spine MRI Protocols—Baseline

Regarding the baseline evaluation of MS patients, the majority of participants acquired a 3D fluid attenuated inversion recovery (FLAIR) volume (40/46, 87.0%), while the remaining opted for a 2D FLAIR sequence (6/46, 13.0%), with a slice thickness superior to 3 mm in almost all cases (5/6, 83.3%). Among the other T2-weighted sequences, the second-most acquired was the 2D T2w sequence (30/46, 65.2%), with a slice thickness superior to 3 mm in half of the cases (15/30, 50.0%), while only few participants (3/46, 6.5%) acquired a 3D T2w volume. Among pre-contrast T1w sequences, 2D spin-echo (SE)-T1w was the most frequently reported (65.2% of the cases), followed by 3D gradient-echo (GrE)-T1w volumes (28.3%). After gadolinium administration, the use of the two sequences was better balanced but still in favor of the 2D acquisition (2D SE-T1w images in 56.5% of the cases and 3D GrE-T1w volumes in 50.0% of the cases). In the majority of cases (28/41, 68.3%), the post-gadolinium sequence was acquired 5 min after the end of contrast administration. A small percentage of participants (5/46, 10.9%) declared to not routinely acquire post-gadolinium sequences at baseline. At baseline, half of the participants routinely acquired a specific sequence for cortical lesions detection (23/46, 50.0%), with the 3D double inversion recovery (DIR) being the preferred choice (14/23, 60.9%). More than 80% of the respondents (38/46, 82.6%) routinely acquired a sequence for optic nerves evaluation, with short tau inversion recovery (STIR)-T2w being the most common choice (34/38, 89.5%). In the framework of optic nerves evaluation, 20 participants also routinely acquired a post-gadolinium fat-saturated T1w sequence (58.8%). Finally, with reference to the spine MRI protocol at baseline, all participants acquired either a T2w or a STIR-T2w sagittal sequence, with a slice thickness inferior or equal to 3 mm in almost all cases (44/46, 95.6%). Sixty-three percent of the participants (29/46) acquired a T1w sequence both before and after contrast administration, while 16 NRs (34.8%) acquired a post-gadolinium sequence only. In almost all cases, spine T1w sequences had a 2D resolution (43/46, 93.5%). Of these, only 23 (60.5%) had a thickness equal or inferior to 3 mm without a gap. Results relative to the baseline MRI protocol are summarized in Table 1 and Figure 2.

### 3.3. Brain and Spine MRI Protocols—Follow-Up

Regarding the brain acquisition at a follow-up examination, it was confirmed that most of the participants acquired a 3D FLAIR sequence (37/46, 80.4%), in line with data obtained from baseline evaluation. Similarly, among the other T2w images as well, the second-most used was the 2D T2w sequence (31/46, 67.4%), with a thickness superior to 3mm in about half cases (15/31, 48.4%). Finally, the use of pre-contrast T1w sequences was also in line with the baseline (SE-T1w: 28/46, 60.9%; 3D GrE-T1w: 10/46, 21.7%). The vast majority of participants (40/46, 87.0%) declared that they perform at least one post-contrast T1w sequence, either a 3D GrE-T1w (23/40, 57.5%) or a 2D SE-T1w sequence (21/40, 52.5%). Compared to the baseline evaluation, the acquisition of specific sequences for cortical lesions detection was slightly less frequent on follow-up examination (20/46 of cases, 43.5%), with the 3D DIR sequence remaining the NRs’ preferred choice (11/20, 55.0%). A reduction in the percentage of participants acquiring a specific sequence for optic nerve evaluation was also observed, with 24/46 NR (52.2%) acquiring a STIR-T2w sequence, followed in less than half cases by contrast administration (11/24, 45.8%). Finally, sequences acquired for spine evaluation at follow-up mirrored the one applied for the baseline evaluation in terms of T2w acquisition (43/46, 93.5%). Interestingly, a similar stable pattern was observed in terms of contrast administration at follow-up, with 40/46 NRs (87.0%) routinely administering Gadolinium for the execution of a post-contrast T1w sequence alone (19/40, 47.5%) or along with a pre-contrast acquisition (21/40, 52.5%). Results relative to the follow-up MRI protocol are summarized in Table 2 and Figure 3.

### 3.4. Stratification By Work-Related Environment

When stratifying results on the basis of the respondents’ workplace, we found that the acquisition of the 3D FLAIR sequence was substantially comparable (private clinics: 8/9, 88.9%; hospitals: 20/23, 87.0%; university hospitals: 12/14, 85.7%). Regarding brain sequences acquired after Gadolinium administration, the 2D SE-T1w was the preferred choice in hospitals (13/21, 61.9%) and university hospitals (8/14, 57.1%), while 3D GrE-T1w was the most acquired sequence in private clinics (6/9, 66.7%). When a 2D SE-T1w was acquired, it was often obtained with a slice thickness superior to 3 mm (16/26, 61.5%), mostly in private clinical centers (4/5, 80.0%) and hospitals (8/13, 61.5%), but also in university hospitals (4/8, 50.0%).

### 3.5. Stratification by Years of Experience

When stratifying data by years of experience, we found that all NRs with less than 10 years of experience usually acquired a 3D FLAIR sequence (22/22, 100.0%), whereas this percentage was reduced to 77.3% (17/22) when evaluating results obtained from NRs with more than 10 years of experience. Regarding the Gadolinium contrast-enhanced sequences, we found that the 2D SE-T1w was the most performed sequence by young NRs (13/18, 72.2%), while the subgroup of more experienced ones acquired both 3D-GrE-T1w and 2D SE-T1w in less than 50% of the cases (9/22, 40.9%). Notably, the 2D SE-T1w was performed with a slice thickness superior to 3 mm in almost all cases by older NRs (8/9, 88.9%). Finally, the report structure also proved to be slightly different between less and more experienced NRs, with a structured report being made by young NRs in half of the cases (4/8, 50.0%).

### 3.6. Report Structure

Regarding the report structure, most of the participants (38/46, 82.6%) routinely preferred a descriptive report over a standardized one. In most of the cases, report drafting required 15 to 30 min per patient (31/46, 67.4%). Only a few NRs (14/46, 30.4%) clearly referred to the 2017 revision of the McDonald criteria for MS diagnosis in their report. In a relatively large number of cases, NRs specified the MRI protocol used (41/46, 89.1%), the anatomical areas covered (35/46, 76.1%), and the contrast agent type and dose (34/46, 73.9%). On the other hand, in less than half of the cases (19/46, 41.3%), the magnetic field intensity was included in the report, and slice thickness was only rarely declared (5/46, 10.9%). Most of participants included in their report information about lesions number (31/46, 67.4%), as a range and not as an exact number, as well as the precise number of contrast enhancing lesions. Regarding lesion localization, almost all participants routinely specify the lesion location (45/46, 97.8%), focusing on only typical MS locations in most cases (25/45, 55.6%) rather than specifying the specific anatomic region affected (20/45, 44.4%). Less than 20% of the participants (8/46, 17.4%) clearly reported the presence of optic nerves involvement, while information about the presence of atrophy (via qualitative evaluation, 39/46, 84.8%), black holes (38/46, 82.6%), and cortical lesions number (37/46, 80.4%) were frequently included. Finally, we observed a substantial correspondence between information provided at baseline and follow-up reports, with information about new lesions (43/46. 93.5%) and their volume increase (42/46, 91.3%) on T2w sequences also provided at follow-up.

### 3.7. Future Directions

Lastly, the participants were asked to select which MS emerging physiopathological marker they believed to be ready for clinical use. Most of them indicated the central vein sign and the evaluation of brain atrophy via a quantitative assessment (in both cases, 24/46, 52.2%) as the two most plausible future diagnostic markers of the disease, while the evaluation of the “slowly expanding lesions” was judged relatively far from a possible use in a clinical setting (6/46, 13%). Nine out of 46 NRs (19.6%) claimed that none of the proposed signs were ready to be translated into clinical practice, mostly because they felt there was a lack of proper preparation to recognize those markers (6/9, 66.7%) or believed technical limitations would hamper the acquisition of specific sequences required for biomarkers assessment (4/9, 44.4%).

## 4. Discussion

The central role of MRI in the evaluation of MS patients is unquestionable [11]. Over the years, MRI has become one of the most important diagnostic tools available to NRs and clinicians, also providing relevant insights about the pathophysiology of damage in MS and offering new possibilities to monitor disease progression and treatment response [12]. In light of the crucial role of MRI in clinical settings [13], expert guidelines have been produced to ensure that MRI protocols for evaluations of MS patients would include sequences adequate for the application of the ever-evolving MS diagnostic criteria and longitudinal monitoring of disease activity. Specifically, the MAGNIMS committee identifies mandatory and optional sequences. Among the mandatory sequences, a particular emphasis is placed on the use of FLAIR-T2w sequences and post-contrast T1w sequences [8]. The results of this study show that although the MS community is somehow close to a standardization of MRI protocols, there is still room for a certain degree of variability in clinical practice. For instance, although most of the participants demonstrated an overall good adherence to the recommendations about post-gadolinium sequences, almost 10% of the participants did not follow the MAGNIMS recommendations, only acquiring 2D sequences with a slice thickness superior to 3 mm. A possible explanation could be the need for reducing acquisition times, although the diagnostic efficacy of such an approach is obviously debatable. In this light, an additional nonadherence to the current guidelines, perhaps even more crucial than the previous one, was that about 30% of NR claimed to acquire the post-contrast T1w sequences within 5 min of the contrast administration. This point is crucial, given that it is well established that lesions peak enhancement occurs 10 min after contrast administration [10,14]. Changes regarding the timing of post-gadolinium T1w acquisition could affect the diagnostic workflow of these patients, given that contrast enhancement allows for determination of DIT at diagnosis [4]. Finally, and still related to gadolinium administration issues, the results of this study suggest a usage of contrast administration not in line with the current guidelines, both at baseline (where almost 10% of participants declared to not routinely acquire post-Gd sequence in cases of suspected MS) and especially at follow-up. In particular, we found a similar percentage of NRs routinely administering a contrast agent both at baseline and follow-up examinations. A possible explanation for this phenomenon could be researched in a historical heritage, as for years, contrast administration was thought to be the only way to demonstrate disease activity and progression, even using double or triple doses [15,16,17]. Nevertheless, it is now well established that disease progression can be assessed by evaluating different MRI parameters, although unfortunately it appears that such information does not translate into a reduced request for and execution of post gadolinium sequences. Beyond the relative utility of contrast agent administration in follow-up studies, recent concerns about brain gadolinium deposition [18,19,20] represent an additional reason to limit the use of gadolinium-based agents to specific cases, although the clinical impact of gadolinium deposition is far from being completely understood [21,22,23]. As per the opportunity to evaluate the presence of CL, not only is cortical involvement considered as typical by the most recent MS diagnostic criteria, but numerous studies have shown that its presence and number is correlated with both motor and cognitive disability in MS patients [24,25,26,27,28] since the early phases of the disease [29]. Accordingly, about 80% of NRs routinely evaluated the presence of CL in MS baseline evaluation, via the application of dedicated sequences in half of the cases. Among these, DIR was the most widely applied in our sample, in line with the increased attention gained by this sequence in recent years [30,31]. It has to be noted that in a very small percentage of cases (less than 5%), NRs answered that they routinely work on a scanner with a field strength inferior to 1.5T, acquiring only standard clinical sequences, such as 2D SE-T1w, 2D T2w, and FLAIR sequences, thus further reducing the possibility to evaluate such lesions in this small subgroup. Regarding the study report, in our sample, NRs preferred a descriptive report over a standardized one. As per other districts and conditions [32,33,34], the use of a standardized structured report in MS is strongly warranted [35,36], given the intrinsic potential of such an approach to improve diagnostic accuracy and reproducibility over time, by solving problems such as ambiguity and lack of completeness that can occur with conventional narrative reports, with an obvious and direct impact on the quality of care. In this light, it was interesting to note that a relatively low percentage of participants usually mention the 2017 revision of the McDonald Criteria [4] in their report. Although it emerged that NRs in almost all cases refer in their report to typical locations needed to achieve DIS, it would be advisable to clearly state the current diagnostic criteria in the report, in line with the MAGNIMS consensus guidelines suggesting that the report must always contain a conclusion to communicate the radiological interpretation in relation to the clinical problem. Finally, participants were asked which, if any, of the emerging imaging markers was deemed to be included in the near future as part of the neuroradiological evaluation of MS in clinical practice. Among all of the imaging biomarkers, more than half of the NRs identified the central vein sign as a promising and useful marker for MS diagnosis, in line with research studies suggesting its application to increase diagnostic sensitivity and specificity [37,38,39,40], followed by a quantitative assessment of cerebral atrophy. As per this latter biomarker, although measures of brain atrophy are valuable markers of ongoing neurodegeneration in MS [41,42], and an atrophy cut-off able to differentiate pathological from physiological atrophy accrual has been identified [43], we must not forget that the determination of meaningful atrophy in the single-subject is still beyond our reach. Interestingly, less confidence in diagnostic sensitivity and specificity was reserved to other neuroradiological biomarkers such as the “slow-expanding lesions”, probably due to methodological concerns. The main limitation of this study is represented by the small cohort of NRs involved. Indeed, although the survey reached a wide range of professional figures of various years of experience and Work-Related Environment, the overall numerosity is unfortunately too low to make assumptions about a more profound generalization of these findings. For this reason, future similar studies are warranted, reaching a larger number of international participants, to confirm the findings reported here. Although characterized by this limitation, in conclusion, this work showed that even if recommendations about standardized MRI protocols are present, in clinical practice, a relatively large degree of heterogeneity is still present regarding the neuroradiological evaluation of MS patients. Further work to overcome this issue is mandatory in the MS community to reduce differences among centers with the final aim to increase the quality of patients’ care in this condition.

## Figures and Tables

**Figure 1 diagnostics-11-01310-f001:**
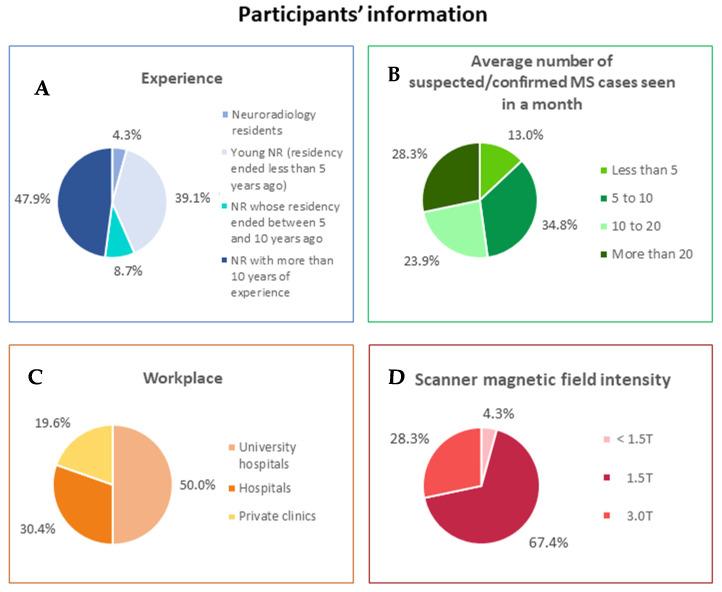
Participants’ information. Pie charts represent (**A**) experience in neuroradiology field; (**B**) average number of suspected/confirmed Multiple Sclerosis cases usually seen in a month; (**C**) workplace; (**D**) scanner magnetic field intensity. Data are show as percentages.

**Figure 2 diagnostics-11-01310-f002:**
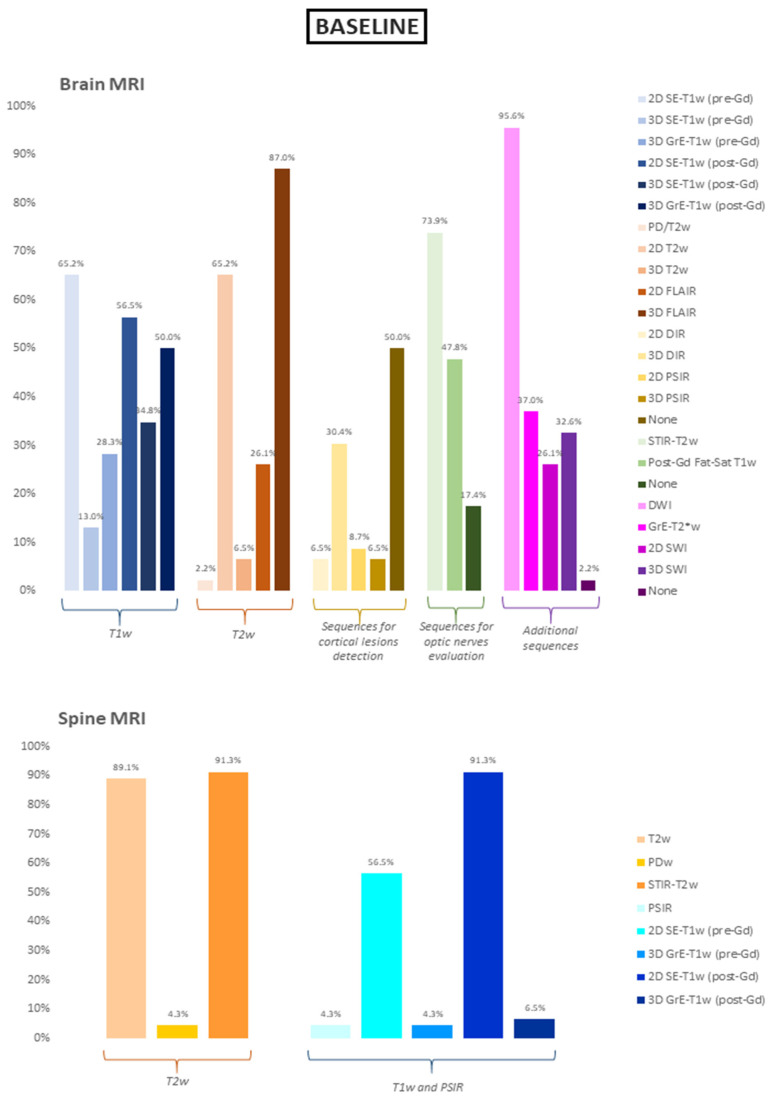
Sequences included in the baseline brain and spine MRI protocols. Data are show as percentages. SE = spin-echo; GrE = gradient-echo; FLAIR = fluid attenuated inversion recovery; DIR = double inversion recovery; PSIR = phase sensitive inversion recovery; Fat-Sat = fat saturated; DWI = diffusion weighted imaging; SWI = susceptibility weighted imaging; PD = proton density; STIR = short tau inversion recovery; Gd = gadolinium.

**Figure 3 diagnostics-11-01310-f003:**
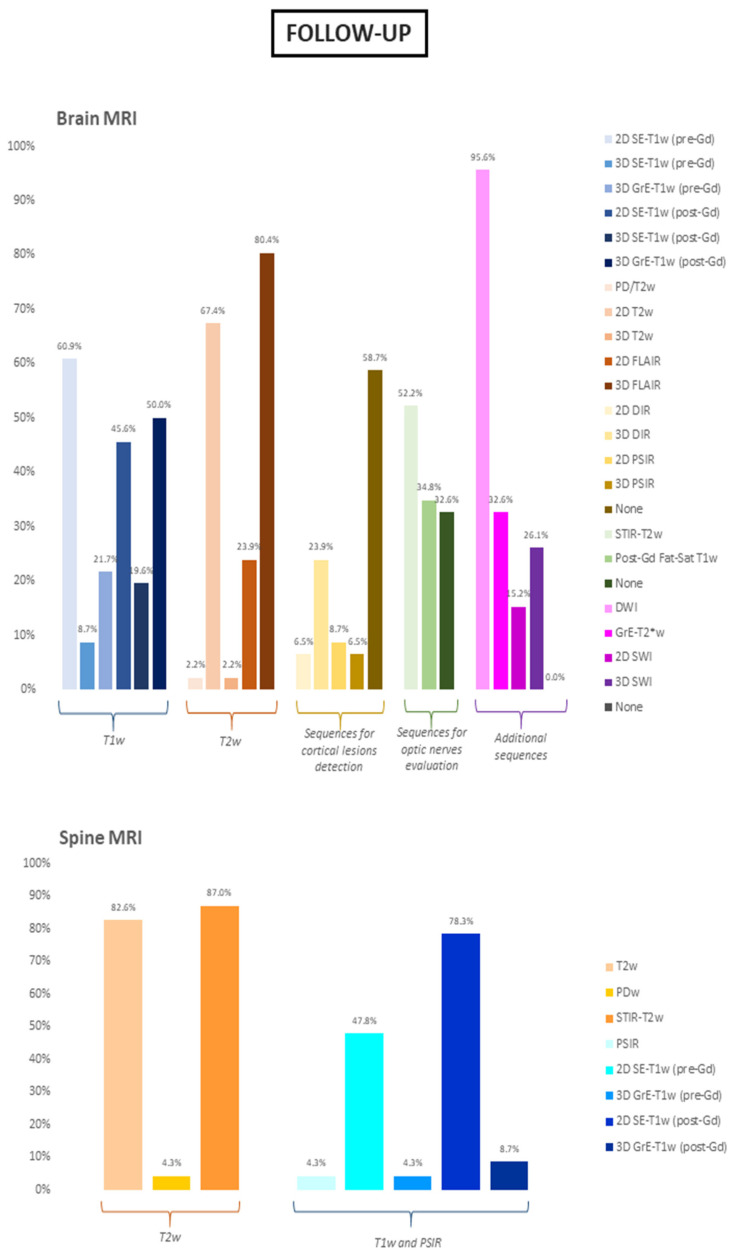
Sequences included in the follow-up brain and spine MRI protocols. Data are show as percentages. SE = spin-echo; GrE = gradient-echo; FLAIR = fluid attenuated inversion recovery; DIR = double inversion recovery; PSIR = phase sensitive inversion recovery; Fat-Sat = fat saturated; DWI = diffusion weighted imaging; SWI = susceptibility weighted imaging; PD = proton density; STIR = short tau inversion recovery; Gd = gadolinium.

**Table 1 diagnostics-11-01310-t001:** Results of the survey regarding brain and spine MRI at baseline.

BASELINE						
BRAIN MRI						
*MRI SEQUENCES*	*Number of Affirmative Answers*	*Percentage*	*Additional Questions*		*Number of Answers*	*Percentage*
***T1-weighted***						
2D SE-T1w (pre-Gd)	30	65.2%	slice thickness	≤3 mm	8	26.7%
				>3 mm	22	73.3%
			gap	Yes	10	33.3%
				No	20	66.7%
3D SE-T1w (pre-Gd)	6	13.0%	voxel size	<1 mm (isotropic)	3	50.0%
				1 mm (isotropic)	3	50.0%
				other	0	0.0%
3D GrE-T1w (pre-Gd)	13	28.3%	voxel size	<1 mm (isotropic)	1	7.7%
				1 mm (isotropic)	12	92.3%
				other	0	0.0%
2D SE-T1w (post-Gd)	26	56.5%	slice thickness	≤3 mm	10	38.5%
				>3 mm	16	61.5%
			gap	Yes	11	42.3%
				No	15	57.7%
			delay	≤5′	11	42.3%
				>5′	15	57.7%
3D SE-T1w (post-Gd)	16	34.8%	voxel size	<1 mm (isotropic)	6	37.5%
				1 mm (isotropic)	8	50.0%
				other	2	12.5%
			delay	≤5′	11	68.7%
				>5′	5	31.3%
3D GrE-T1w (post-Gd)	23	50.0%	voxel size	<1 mm (isotropic)	3	13.6%
				1 mm (isotropic)	18	81.8%
				other	1	4.6%
			delay	≤5′	7	31.8%
				>5′	15	68.2%
***T2-weighted***						
PD/T2w	1	2.2%	slice thickness	≤3 mm	1	100%
				>3 mm	0	0.0%
			gap	Yes	0	0.0%
				No	1	100.0%
2D T2w	30	65.2%	slice thickness	≤3 mm	15	50.0%
				>3 mm	15	50.0%
			gap	Yes	13	43.3%
				No	17	56.7%
3D T2w	3	6.5%	voxel size	<1 mm (isotropic)	1	33.3%
				1 mm (isotropic)	1	33.3%
				other	1	33.3%
2D FLAIR	12	26.1%	slice thickness	≤3 mm	4	33.3%
				>3 mm	8	66.7%
			gap	Yes	2	16.7%
				No	10	83.3%
3D FLAIR	40	87.0%	voxel size	<1 mm (isotropic)	10	25.0%
				1 mm (isotropic)	24	60.0%
				other	6	15.0%
***Sequences for cortical lesions detection***						
2D DIR	3	6.5%	slice thickness	≤3 mm	2	66.7%
				>3 mm	1	33.3%
			gap	Yes	1	33.3%
				No	2	66.7%
3D DIR	14	30.4%	voxel size	<1 mm (isotropic)	2	14.3%
				1 mm (isotropic)	10	71.4%
				other	2	14.3%
2D PSIR	4	8.7%	slice thickness	≤3 mm	2	50.0%
				>3 mm	2	50.0%
			gap	Yes	2	50.0%
				No	2	50.0%
3D PSIR	3	6.5%	voxel size	<1 mm (isotropic)	0	0.0%
				1 mm (isotropic)	3	100%
				other	0	0.0%
None	23	50.0%				
***Sequences for optic nerves evaluation***						
STIR-T2w	34	73.9%	slice thickness	≤3 mm	31	91.2%
				>3 mm	3	8.8%
			gap	Yes	10	29.4%
				No	24	70.6%
Post-Gd Fat-Sat T1w	22	47.8%	slice thickness	≤3 mm	22	100%
				>3 mm	0	0.0%
			gap	Yes	5	22.7%
				No	17	77.3%
None	8	17.4%				
***Additional sequences***						
DWI	44	95.6%				
GrE-T2*	17	37.0%	slice thickness	≤3 mm	4	23.5%
				>3 mm	13	76.5%
			gap	Yes	8	47.1%
				No	9	53.9%
2D SWI	12	26.1%	slice thickness	≤3 mm	9	75.0%
				>3 mm	3	25.0%
			gap	Yes	2	16.7%
				No	10	83.3%
3D SWI	15	32.6%	voxel size	<1 mm (isotropic)	4	26.7%
				1 mm (isotropic)	9	60.0%
				other	2	13.3%
None	1	2.2%				
**SPINE MRI**						
***MRI SEQUENCES***	***Number of Affirmative Answers***	***Percentage***	***Additional Questions***		***Number of Answers***	***Percentage***
***T2-weighted***						
T2w	41	89.1%	acquisition plane	sagittal	41	100%
				axial	17	41.5%
			slice thickness on sagittal acquisition	≤3 mm	39	95.1%
				>3 mm	2	4.9%
			gap on sagittal acquisition	Yes	14	34.1%
				No	27	65.9%
PDw	2	4.3%	acquisition plane	sagittal	2	100.0%
				axial	0	0.0%
			slice thickness on sagittal acquisition	≤3 mm	1	50.0%
				>3 mm	1	50.0%
			gap on sagittal acquisition	Yes	2	100.0%
				No	0	0.0%
STIR-T2w	42	91.3%	acquisition plane	sagittal	42	100.0%
				axial	2	4.7%
			slice thickness on sagittal acquisition	≤3 mm	38	90.5%
				>3 mm	4	9.5%
			gap on sagittal acquisition	Yes	14	33.3%
				No	28	66.7%
***T1-weighted and inversion recovery***						
PSIR	2	4.3%	acquisition plane	sagittal	2	100%
				axial	0	0.0%
			slice thickness on sagittal acquisition	≤3 mm	2	100%
				>3 mm	0	0.0%
			gap on sagittal acquisition	Yes	1	50.0%
				No	1	50.0%
2D SE-T1w (pre-Gd)	26	56.5%	acquisition plane	sagittal	26	100%
				axial	3	11.5%
			slice thickness on sagittal acquisition	≤3 mm	23	88.5%
				>3 mm	3	11.5%
			gap on sagittal acquisition	Yes	11	42.3%
				No	15	57.7%
3D GrE-T1w (pre-Gd)	2	4.3%	acquisition plane	sagittal	2	100%
				axial	1	50.0%
			slice thickness on sagittal acquisition	≤3 mm	2	100%
				>3 mm	0	0.0%
			gap on sagittal acquisition	Yes	0	0.0%
				No	2	100%
2D SE-T1w (post-Gd)	42	91.3%	acquisition plane	sagittal	41	100%
				axial	13	31.7%
			slice thickness on sagittal acquisition	≤3 mm	37	90.2%
				>3 mm	4	9.8%
			gap on sagittal acquisition	Yes	16	39.0%
				No	25	61.0%
			delay	≤5′	19	47.5%
				>5′	21	52.5%
3D GrE-T1w (post-Gd)	3	6.5%	acquisition plane	sagittal	2	66.7%
				axial	2	66.7%
			slice thickness on sagittal acquisition	≤3 mm	2	100%
				>3 mm	0	0.0%
			gap on sagittal acquisition	Yes	0	0.0%
				No	2	100%
			delay	≤5′	1	33.3%
				>5′	2	66.7%

SE = spin-echo; GrE = gradient-echo; FLAIR = fluid attenuated inversion recovery; DIR = double inversion recovery; PSIR = phase sensitive inversion recovery; Fat-Sat = fat saturated; DWI = diffusion weighted imaging; SWI = susceptibility weighted imaging; PD = proton density; STIR = short tau inversion recovery; Gd = gadolinium.

**Table 2 diagnostics-11-01310-t002:** Results of the survey regarding brain and spine MRI at follow-up.

FOLLOW-UP						
BRAIN MRI						
*MRI SEQUENCES*	*Number of Affirmative Answers*	*Percentage*	*Additional Questions*		*Number of Answers*	*Percentage*
***T1-weighted***						
2D SE-T1w (pre-Gd)	28	60.9%	slice thickness	≤3 mm	10	35.7%
				>3 mm	18	64.3%
			gap	Yes	12	42.9%
				No	16	57.1%
3D SE-T1w (pre-Gd)	4	8.7%	voxel size	<1 mm (isotropic)	2	66.7%
				1 mm (isotropic)	1	33.3%
				other	0	0.0%
3D GrE-T1w (pre-Gd)	10	21.7%	voxel size	<1 mm (isotropic)	1	10.0%
				1 mm (isotropic)	9	90.0%
				other	0	0.0%
2D SE-T1w (post-Gd)	21	45.6%	slice thickness	≤3 mm	9	42.9%
				>3 mm	12	57.1%
			gap	Yes	5	23.8%
				No	16	76.2%
			delay	≤5′	12	57.1%
				>5′	8	38.1%
3D SE-T1w (post-Gd)	9	19.6%	voxel size	<1 mm (isotropic)	3	33.3%
				1 mm (isotropic)	5	55.5%
				other	1	11.1%
			delay	≤5′	6	66.6%
				>5′	3	33.3%
3D GrE-T1w (post-Gd)	23	50.0%	voxel size	<1 mm (isotropic)	3	13.0%
				1 mm (isotropic)	18	78.3%
				other	0	0.0%
			delay	≤5′	5	21.7%
				>5′	16	70.0%
***T2-weighted***						
PD/T2w	1	2.2%	slice thickness	≤3 mm	0	0.0%
				>3 mm	1	100%
			gap	Yes	0	0.0%
				No	1	100%
2D T2w	31	67.4%	slice thickness	≤3 mm	14	45.2%
				>3 mm	15	48.4%
			gap	Yes	11	35.5%
				No	18	58.1%
3D T2w	1	2.2%	voxel size	<1 mm (isotropic)	1	100%
				1 mm (isotropic)	0	0.0%
				other	0	0.0%
2D FLAIR	11	23.9%	slice thickness	≤3 mm	3	27.3%
				>3 mm	8	72.7%
			gap	Yes	2	18.2%
				No	9	81.8%
3D FLAIR	37	80.4%	voxel size	<1 mm (isotropic)	9	24.3%
				1 mm (isotropic)	23	62.2%
				other	5	13.5%
***Sequences for cortical lesions detection***						
2D DIR	3	6.5%	slice thickness	≤3 mm	3	100%
				>3 mm	0	0.0%
			gap	Yes	0	0.0%
				No	3	100%
3D DIR	11	23.9%	voxel size	<1 mm (isotropic)	3	27.3%
				1 mm (isotropic)	7	63.6%
				other	1	9.1%
2D PSIR	4	8.7%	slice thickness	≤3 mm	2	50.0%
				>3 mm	1	25.0%
			gap	Yes	2	50.0%
				No	1	25.0%
3D PSIR	3	6.5%	voxel size	<1 mm (isotropic)	1	33.3%
				1 mm (isotropic)	2	66.7%
				other	0	0.0%
None	27	58.7%				
***Sequences for optic nerves evaluation***						
STIR-T2w	24	52.2%	slice thickness	≤3 mm	22	91.7%
				>3 mm	2	8.3%
			gap	Yes	6	25.0%
				No	18	75.0%
Post-Gd Fat-Sat T1w	16	34.8%	slice thickness	≤3 mm	16	100%
				>3 mm	0	0.0%
			gap	Yes	4	25.0%
				No	12	75.0%
None	15	32.6%				
***Additional sequences***						
DWI	44	95.6%				
GrE-T2*	15	32.6%	slice thickness	≤3 mm	4	26.7%
				>3 mm	11	73.3%
			gap	Yes	6	40.0%
				No	9	60.0%
2D SWI	7	15.2%	slice thickness	≤3 mm	5	71.4%
				>3 mm	2	28.6%
			gap	Yes	2	28.6%
				No	5	71.4%
3D SWI	12	26.1%	voxel size	<1 mm (isotropic)	3	25.0%
				1 mm (isotropic)	9	75.0%
				other	0	0.0%
None	0	0.0%				
**SPINE MRI**						
***MRI SEQUENCES***	***Number of Affirmative Answers***	***Percentage***	***Additional Questions***		***Number of Answers***	***Percentage***
***T2-weighted***						
T2w	38	82.6%	acquisition plane	sagittal	38	100%
				axial	12	31.6%
			slice thickness on sagittal acquisition	≤3 mm	37	97.4%
				>3 mm	1	2.6%
			gap on sagittal acquisition	Yes	14	36.8%
				No	24	63.2%
PDw	2	4.3%	acquisition plane	sagittal	2	100%
				axial	0	0.0%
			slice thickness on sagittal acquisition	≤3 mm	2	100%
				>3 mm	0	0.0%
			gap on sagittal acquisition	Yes	2	100%
				No	0	0.0%
STIR-T2w	40	87.0%	acquisition plane	sagittal	40	100%
				axial	5	12.5%
			slice thickness on sagittal acquisition	≤3 mm	38	95.0%
				>3 mm	2	5.0%
			gap on sagittal acquisition	Yes	16	40.0%
				No	24	60.0%
***T1-weighted and inversion recovery***						
PSIR	2	4.3%	acquisition plane	sagittal	2	100%
				axial	0	0.0%
			slice thickness on sagittal acquisition	≤3 mm	2	100%
				>3 mm	0	0.0%
			gap on sagittal acquisition	Yes	1	50.0%
				No	1	50.0%
2D SE-T1w (pre-Gd)	22	47.8%	acquisition plane	sagittal	22	100%
				axial	5	22.7%
			slice thickness on sagittal acquisition	≤3 mm	22	100%
				>3 mm	0	0.0%
			gap on sagittal acquisition	Yes	8	36.4%
				No	14	63.6%
3D GrE-T1w (pre-Gd)	2	4.3%	acquisition plane	sagittal	2	100%
				axial	1	50.0%
			slice thickness on sagittal acquisition	≤3 mm	2	100%
				>3 mm	0	0.0%
			gap on sagittal acquisition	Yes	0	0.0%
				No	2	100%
2D SE-T1w (post-Gd)	36	78.3%	acquisition plane	sagittal	36	100%
				axial	9	25.0%
			slice thickness on sagittal acquisition	≤3 mm	33	91.6%
				>3 mm	2	5.5%
			gap on sagittal acquisition	Yes	13	36.1%
				No	22	61.1%
			delay	≤5′	17	47.2%
				>5′	19	52.8%
3D GrE-T1w (post-Gd)	4	8.7%	acquisition plane	sagittal	4	100%
				axial	1	25.0%
			slice thickness on sagittal acquisition	≤3 mm	3	75.0%
				>3 mm	1	25.0%
			gap on sagittal acquisition	Yes	1	25.0%
				No	3	75.0%
			delay	≤5′	0	0.0%
				>5′	4	100%

SE = spin-echo; GrE = gradient-echo; FLAIR = fluid attenuated inversion recovery; DIR = double inversion recovery; PSIR = phase sensitive inversion recovery; Fat-Sat = fat saturated; DWI = diffusion weighted imaging; SWI = susceptibility weighted imaging; PD = proton density; STIR = short tau inversion recovery; Gd = gadolinium.

## Data Availability

Data will be made available upon reasonable request to the corresponding author.

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
