# Peer review of "Walk Your Talk: Real-World Adherence to Guidelines on the Use of MRI in Multiple Sclerosis"

_diagnostics, 2021, doi:10.3390/diagnostics11081310_

Round 1

Reviewer 1 Report

In this article, Tortora et al. investigated the different imaging prototocols as well as the opinion of emerging imaging markers in a real life setting using an online survey. In sum, the data are of interest, as they reveal the situation of MS-imaging analyes in an italian real life situation.

I do have some minor points:

Introduction:

  • line 64: ...the McDonald criteria are based on ... using MRI. - please clarify. DIS and DIT can be based on MRI, but there are also clinical or CSF markers to fulfil DIS and DIT.

Materials:

  • It would be intersting to know how many NRs have been asked. Or did the autohrs only ask this 46NR? What was the way the NRs have been recruited?

Results:

  • line 116: 37/46, not 37/64
  • line 129: Please explain the abbreation at the time of the first mention (FLAIR, SE, GrE, DIR...)
  • it would also be interested to have a more detailed view if for example the high experienced NR, or the NR with lots of cases per month, in the way 3.4. has been reported; and also if the report structure is depending on experience, cases or institution

discussion:

  • the most influencing factors are time, and the amount of money the NR will get. I do not know the invoice or MRI scans in italy, but maybe that might also be discussed.
  • a few scanners have been below 1.5T field strength. This might also influece the protocoll? I dont think that DIR sequences are available at these scanners.

In sum the data are of some interest for the scientific community, so I recommend minor revision.

Author Response

Reviewer #1

In this article, Tortora et al. investigated the different imaging prototocols as well as the opinion of emerging imaging markers in a real life setting using an online survey. In sum, the data are of interest, as they reveal the situation of MS-imaging analyes in an italian real life situation.

I do have some minor points:

Introduction:

line 64: ...the McDonald criteria are based on ... using MRI. - please clarify. DIS and DIT can be based on MRI, but there are also clinical or CSF markers to fulfil DIS and DIT.

We thank the Referee for the valuable suggestion, and we have modified the Manuscript accordingly (Page 2, Lines 65-66)

Materials:

It would be intersting to know how many NRs have been asked. Or did the autohrs only ask this 46NR? What was the way the NRs have been recruited?

We thank the Referee for giving us the possibility to clarify the methodology adopted.

Based on our connections, we have reached at least three neuroradiologist (one for each setting explored, i.e. university hospital, hospital and clinical practice) per each Italian region, in an attempt to enroll a sample representative of the entire country. We have then asked each reached NR who showed interest in participating to this work to disseminate the survey to colleagues working in different centers, to increase the sample size.    

This is now specified on Page 3, Lines 83-87 of the revised version of the Manuscript.

Results:

line 116: 37/46, not 37/64

We apologize for the typo, and modified the Manuscript accordingly (Page 4, Line 120).

line 129: Please explain the abbreation at the time of the first mention (FLAIR, SE, GrE, DIR...)

We thank the Referee for the suggestion and following also the concern #6 raised by Reviewer #2, we have spelled-out all the abbreviations included in the text in their first appearance.

it would also be interested to have a more detailed view if for example the high experienced NR, or the NR with lots of cases per month, in the way 3.4. has been reported; and also if the report structure is depending on experience, cases or institution

We thank the Reviewer for this interesting suggestion. We have now added a new subparagraph to the Results Section (3.5 Stratification by years of experience), in which results have been stratified by years of experience. (Page 17, Lines 221-231), as well as  introducing this analysis in the Methods (Page 3, Line 111)

discussion:

the most influencing factors are time, and the amount of money the NR will get. I do not know the invoice or MRI scans in italy, but maybe that might also be discussed.

We thank the Reviewer for giving us the possibility to clarify this point.

Although the increasing demand for cost-effectiveness and efficiency in hospitals is for sure crucial, NRs working in Italian hospital and university hospital receive a fixed monthly salary that is independent from the amount of reports written. On the other hand, NRs working in clinical practice usually receive a compensation based on the amount of reports written, although this is not universally true given that some centers allow for a fixed salary also independent from the amount of exams evaluated.

Given this background, we have the feeling that discussing this point might not add any significant point of interest to our work. We hope that this will not create a major setback for the publication of our Work.

a few scanners have been below 1.5T field strength. This might also influece the protocoll? I dont think that DIR sequences are available at these scanners.

We agree with the Referee that some sequence, such as DIR or isotropic T1-weighted volumes, might not be available on scanners below 1.5T of field strength. However, considering that less than 5% of responders worked on scanners with field strength lower than 1.5T, we feel that reporting these results separately would not yield any meaningful information, nor removing them from the overall results would modify significantly the reported percentages. We have nevertheless added a sentence in the Discussion section to comment this point (Page 19, Lines 314-318)

We hope this clarifies

In sum the data are of some interest for the scientific community, so I recommend minor revision.

We thank again the Referee for the suggestions that have been made.

Reviewer 2 Report

Summary

Aim of this study was to evaluate the real-world application of MS imaging guidelines in different settings, to clarify the level of adherence to these guidelines. By adopting an on-line anonymous survey, neuroradiologists (NR) were asked about MRI protocols and parameters routinely acquired when MS patients are evaluated in their center, both at diagnosis and follow-up. Forty-six participants were included, mostly working in a hospital or university hospital (80.4%) and with more than 10 years of experience (47.9%). The Authors found a relatively good adherence to the suggested MRI protocols regarding the use of T2-weighted sequences, while a wider degree of heterogeneity was found regarding gadolinium administration.

Comments to the authors

The study aim is interesting few minor comments are reported below.

Introduction

  • As MS also causes GM damage, the first sentence needs to be replaced.

Methods

  • Did the Authors take into account within-center heterogeneity? If yes, How?
  • Residents should not be included as they usually do not participate to MRI protocol definition.
  •  

Results

  • Do the scanner intensity field affect the MRI protocol adopted? This point should be specified.
  • It could be interesting to observe any factors influencing the variability observed.
  • It could be interesting to explore the reasons underlying neuroradiologist choices in MRI acquisition protocol definition.

Discussion

  • All the abbreviations adopted need to be spelled-out.

Author Response

Reviewer #2

Summary

Aim of this study was to evaluate the real-world application of MS imaging guidelines in different settings, to clarify the level of adherence to these guidelines. By adopting an on-line anonymous survey, neuroradiologists (NR) were asked about MRI protocols and parameters routinely acquired when MS patients are evaluated in their center, both at diagnosis and follow-up. Forty-six participants were included, mostly working in a hospital or university hospital (80.4%) and with more than 10 years of experience (47.9%). The Authors found a relatively good adherence to the suggested MRI protocols regarding the use of T2-weighted sequences, while a wider degree of heterogeneity was found regarding gadolinium administration.

Comments to the authors

The study aim is interesting few minor comments are reported below.

Introduction

As MS also causes GM damage, the first sentence needs to be replaced.

We thank the Referee for the valuable suggestion, and we have modified the Manuscript accordingly (Page 2, Lines 56-57)

Methods

Did the Authors take into account within-center heterogeneity? If yes, How?

We thank the Referee for pointing out this missing information of our Manuscript. As it is now clearly stated in the Manuscript (Page 3, Lines 83-87), also following the concern raised by Reviewer #1, Point #2, NRs were asked to disseminate the Survey to colleagues of different centers, to avoid redundancy in the results and problems in analyzing and interpreting within-centers data.

We hope this is now better clarified.

Residents should not be included as they usually do not participate to MRI protocol definition.

We thank the Reviewer for this comment.

Although we agree that residents usually do not participate to MRI protocol definition, we here asked participants with a wide range of expertise, including residents, the protocol that are usually acquired in their centers, not the one that they have personally defined on the scanner. Including residents in this survey gave us the indirect opportunity of reaching more centers and retrieve additional information from university hospitals, given that residents by definition only work in such environments.

We hope this will clarify.

Results

Do the scanner intensity field affect the MRI protocol adopted? This point should be specified.

Please see answer to Reviewer #1, Point #7.

It could be interesting to observe any factors influencing the variability observed.

It could be interesting to explore the reasons underlying neuroradiologist choices in MRI acquisition protocol definition.

We agree with the Referee that investigating the factors influencing the variability here observed, as well a exploring and understanding the reasons behind the choice of each center in acquiring a specific sequence rather than another. Nevertheless, as stated in the limitations of the Discussion section, the small cohort here reported, as well as the study design (structured survey not including questions on the reason behind protocol definition), limit our ability to further investigate more profound aspects of these findings, which will be hopefully investigated in future studies including larger groups of participants.

We hope this clarifies.

Discussion

All the abbreviations adopted need to be spelled-out.

Please see answer to Reviewer #1, Point #4.

Round 2

Reviewer 2 Report

No further comments.